# Computational Investigation of Chirality-Based Separation of Carbon Nanotubes Using Tripeptide Library

**DOI:** 10.3390/biom13010175

**Published:** 2023-01-13

**Authors:** Shrishti Singh, Heena R. Divecha, Abimbola Ayoola, Marvin Xavierselvan, Jack Devlin, Isaac Macwan

**Affiliations:** 1Department of Bioengineering, George Mason University, Fairfax, VA 22030, USA; 2Department of Biomedical Engineering, University of Bridgeport, Bridgeport, CT 06604, USA; 3Department of Electrical and Biomedical Engineering, Fairfield University, Fairfield, CT 06834, USA

**Keywords:** carbon nanotubes, CNT sorting, chirality, molecular dynamics, tripeptides

## Abstract

Carbon nanotubes (CNT) have fascinating applications in flexible electronics, biosensors, and energy storage devices, and are classified as metallic or semiconducting based on their chirality. Semiconducting CNTs have been teased as a new material for building blocks in electronic devices, owing to their band gap resembling silicon. However, CNTs must be sorted into metallic and semiconducting for such applications. Formerly, gel chromatography, ultracentrifugation, size exclusion chromatography, and phage display libraries were utilized for sorting CNTs. Nevertheless, these techniques are either expensive or have poor efficiency. In this study, we utilize a novel technique of using a library of nine tripeptides with glycine as a central residue to study the effect of flanking residues for large-scale separation of CNTs. Through molecular dynamics, we found that the tripeptide combinations with threonine as one of the flanking residues have a high affinity for metallic CNTs, whereas those with flanking residues having uncharged and negatively charged polar groups show selectivity towards semiconducting CNTs. Furthermore, the role of interfacial water molecules and the ability of the tripeptides to form hydrogen bonds play a crucial role in sorting the CNTs. It is envisaged that CNTs can be sorted based on their chirality-selective interaction affinity to tripeptides.

## 1. Introduction

Since their discovery in 1991 [1], carbon nanotubes (CNTs) have attracted tremendous attention due to their properties including flexibility, resilience, and electronic behavior [2]. These outstanding properties have made them promising materials for new applications, such as nanoelectronic devices, fuel storage materials, and energy storage devices [3,4]. CNTs can be classified based on the number of walls, which are categorized as single-walled or multi-walled, the latter consisting of nested shells of the former [5]. Based on the chiral vectors, ‘m’ and ‘n’, CNTs can be metallic (difference between chiral vectors being zero or a multiple of three) or semiconducting (difference between chiral vectors not equal to three) and thus can have different electronic properties. These electronic properties of CNTs can be exploited potentially at the molecular scale as components of nanoelectronic devices due to the band gap of semiconducting CNT resembling that of silicon [6]. Since CNTs exist in many combinations with different properties, it is important to sort them in order to utilize their potential for the desired application [7]. One of the early methods used dielectrophoresis for separating and collecting metallic CNTs towards the electrode, leaving semiconducting CNTs in the solvent. Nevertheless, this method has a low separation efficiency [8]. DNA-based ion exchange chromatography was able to sort CNTs with up to 90% purity but had lower yield of chiral tubes (m, n) [9]. Other techniques, such as density gradient ultracentrifugation (DGU), used density difference of (m, n) CNTs in a gradient medium for separation [10]. Simple gel chromatography and aqueous two-phase extraction methods were built and expanded on the DGU sorting phenomenon [11,12]. However, the outcomes from these methods had poor efficiency and yield. Hence, there is a need for a proper scalable and efficient solution to separate the synthesized CNTs.

Furthermore, the applications of CNTs in biotechnology [13] and biosensing [14] require an understanding of interactions of CNTs with heterogeneous biological macromolecules, such as proteins. Detailed theoretical analysis of interactions between CNTs and fundamental biological molecules, such as amino acid residues are few [15]. Additionally, there are few reports on the immobilization of small organic molecules [16,17], such as oligonucleotides [18,19], which does not provide a complete picture on the type of interactions at the interface of CNTs and biological macromolecules. Studying these interactions is crucial because amino acids are the fundamental building blocks of peptides and proteins [15], and it is through the sequence and conformation of these residues that proteins exhibit their functions. Previous studies have used phage display libraries with random peptide sequences [5,20,21] to understand the role of peptides in CNT de-bundling, while our previous work has shown the interactions between the D3 domain of R-type flagellin protein and single-walled carbon nanotubes, where specific interactions between glycine in the D3 domain and metallic CNTs were observed [22].

In this study, the modelling of a tripeptide library from the D3 domain of flagellin, the specific interactions of these tripeptides with both metallic (12, 12) and semiconducting (5, 15) CNTs, and their quantification based on the selectivity of tripeptides is performed using molecular dynamics (MD). The driving forces behind such interactions are largely Van Der Waals and hydrophobic interactions [20,21,22,23,24,25,26]. MD simulations of nine tripeptides, each with a different combination of flanking residues, were used to develop an optimal tripeptide sequence library to differentiate between the two forms of CNTs. The specific role of individual amino acids was identified in the tripeptides during their selective interactions with either metallic (12, 12) or semiconducting CNTs (5, 15) used in this study. Each tripeptide in the library contained a middle glycine residue as a common unit flanked by different amino acids on either side. The role of glycine in its selectivity for metallic CNTs [22] is explained based on the effects of the flanking residues. The data analysis shows that the tripeptide library used in this study has huge implications in differentiating the semiconducting and metallic CNTs based on the interactions with the flanking residues. Furthermore, not only glycine but also the flanking residues play a crucial role in sorting the CNTs. This study has importance and ramifications in developing an easier method of de-bundling and separating CNTs when present in a mixture. It eliminates the need of developing a phage library of random peptides to study peptide–CNT interactions. Figure 1 shows the basic scheme of the nine tripeptides used in this study.

## 2. Materials and Methods

Interactive forces between tripeptides and the two types of CNTs were analyzed using visual molecular dynamics (VMD) [27] graphics program. The simulations were carried out using nanoscale molecular dynamics (NAMD) program [28]. The biophysical system consisting of the tripeptides from the R-type flagellin protein [29] and two different chirality CNTs, metallic (12,12) and semiconducting (5,15), with the dimensions (d × l) of 17 Å × 48 Å and 15 Å × 60 Å, respectively, was modelled using a built-in nanotube builder plugin in VMD. The protein data bank (pdb) file of the flagellin protein (1UCU) was scanned to extract the tripeptides, with glycine as the middle residue and nine individual tripeptide combinations having a variety of flanking residues were chosen as shown in the Appendix A. The individual amino acids that make up the selected tripeptides are listed in the Appendix A with their solubility and charge. 

Separate all-atom simulations were carried out for the two forms of CNTs for a period of 100 ns each, totaling eighteen 100 ns simulations. Each simulation contained one of tripeptides, one of the CNTs, and ~12,500 atoms of water. All simulations used the CHARMM force field [30] and TIP3P water model [31] with a neutralizing salt concentration of NaCl for effective polarization of the water molecules. An Intel Core i9 cluster with a total of 36 cores, and NVIDIA GeForce RTX 2080 GPU was used to perform the simulations.

The temperature in each simulation was maintained at 300 K by Langevin thermostat and a pressure of 1 atm was maintained through Nosé-Hoover Langevin-Piston barostat with an oscillation period of 100 fs and a decay rate of 50 fs assuming periodic boundary conditions. A 5000-step energy minimization of the entire system was first performed for all the tripeptide combinations to reach a minimum potential energy and then equilibrated for 500,000 steps (1 ns). All simulations employed an integrated time step of 2 fs with a cutoff of 10–12 Å for both short-range forces and long-range forces. Particle mesh Ewald algorithm was used to calculate the long-range forces. Root-mean-square deviation (RMSD), hydrogen bonds and NAMD energy extensions from VMD were used to determine the stability, interaction energies between molecules, conformational energies of angles, and dihedrals and interaction energies between CNTs and tripeptides.

Furthermore, TCL scripting was employed to determine the interaction energy of the sidechains, contact area, RMSD of individual residues on different tripeptides, average number of water molecules, and center of mass deviations based on the adsorption distance criteria of 5 Å from the surface of CNTs. Lennard Jones potential was used to determine the soft repulsive and attractive interactions between the tripeptides and CNTs. 

## 3. Results

To understand the interactions between the different combinations of tripeptides and CNTs and to study the role of each flanking residue of glycine during such interactions, data from eighteen 100 ns simulations (nine for metallic (12, 12) CNT and nine for semiconducting (5, 15) CNT) were analyzed and compared. To validate the extracted data from the different NAMD trajectories and to confirm the stability of the complexes, data analysis was performed at the molecular level, amino acid sidechain level, and atomic level. This included analysis of the RMSD of the tripeptides and individual amino acid residues, non-binding interaction energies of the tripeptides as a whole and individual sidechains, contact area of tripeptides, conformational energies in terms of changes in the angle and dihedral energies of the tripeptides, number of interfacial water molecules, and the role of hydrogen bonds and deviations in the center of mass for individual tripeptides. This provided clear and compelling reasoning about the nature of the interface between these tripeptides and CNTs, both metallic and semiconducting. To make such an extensive analysis feasible, the nine combinations of tripeptides were categorized into two groups with one group (Group I) containing at least one flanking residue having a hydrophobic sidechain and the second group (Group II) containing flanking residues with non-hydrophobic sidechains (Figure 1). For each combination of tripeptide, all data are presented in terms of an average value over 100 ns after the stabilization of RMSD trajectory. The first and last frame screenshots of all the eighteen simulation runs are shown in Appendix A.

The results for the RMSD and interaction energy analysis for the two groups of tripeptide combinations and CNTs are shown in Figure 2. Average RMSD, which is a measure of the distance between a list of two-paired points (tripeptide and CNT) over a period of 100 ns was calculated for each combination within the two groups. The deviations in the RMSD data typically indicate the conformational changes and determine the stability of the complexes over a period. As seen from Figure 2A,B, the average RMSD indicates that the tripeptide combinations, ‘GGA’, ‘DGY’, and ‘TGK’ are more stable for metallic CNTs whereas ‘GGL’, ‘TGL’, ‘TGY’, ‘TGG’, and ‘NGE’ show more stability for semiconducting CNTs. Tripeptide ‘DGD’ is an exception showing equal stability for both types of CNTs. However, coupled with the overall interaction energies between the tripeptides and the respective CNTs, it provides a different perspective on the interface as seen from Figure 2C,D. The non-bonding energies for the tripeptide combinations of ‘TGL’, ‘DGY’, ‘TGY’, and ‘TGK’ show more interactions with metallic CNTs, whereas the combinations ‘GGA’, ‘GGL’, ‘TGG’, and ‘NGE’ show better interactions with semiconducting CNTs with ‘DGD’ showing similar interactions for both metallic and semiconducting CNTs. 

Next, we investigated whether the carbon backbone of the tripeptide or the sidechains of these flanking residues is responsible for such affinity. A separate analysis was performed only on the sidechains of the nine tripeptides and the average RMSD of the sidechains were compared with the interaction energy. From Figure 3A,B, the average RMSD of the sidechains indicate that the tripeptides, ‘GGA’, ‘GGL’, and ‘DGY’ show stability for metallic CNTs while ‘TGL’, ‘TGY’, ‘TGG’, ‘DGD’, ‘NGE’, and ‘TGK’ show stability towards semiconducting CNTs. The results for the interaction energy between the tripeptide sidechains and the CNTs is shown in Figure 3C,D. Based on these results, the sidechains of the tripeptides ‘TGL’, ‘DGY’, ‘TGY’, and ‘TGK’ show more interactions with metallic CNTs whereas the sidechains of tripeptides ‘GGA’, ‘GGL’, ‘TGG’, ‘DGD’, and ‘NGE’ display higher interactions with semiconducting CNTs.

It is important to study the role of individual amino acids in the tripeptide in the presence of metallic and semiconducting CNTs. Hence, we selected the atoms of the middle glycine residue and sidechains of flanking residues for individual tripeptides, and their interactions with metallic and semiconducting CNTs were analyzed. We computed the average RMSD for the individual amino acids in the tripeptide and the results are shown in Figure 4A,B. Based on these results, the individual amino acids of the tripeptides ‘GGA’, ‘GGL’, ‘TGL’, ‘DGY’, ‘TGY’, and ‘DGD’ had high stability towards semiconducting CNT, whereas ‘TGG’, ‘NGE’, and ‘TGK’ preferred metallic CNTs. Additionally, we computed the contact area of the tripeptides since the area determines the impact that the tripeptides have for sorting a particular CNT. From Figure 4C,D, the tripeptides ‘TGL’, ‘DGY’, ‘TGY’, and ‘TGK’ had higher contact for metallic CNTs, while ‘GGA’, ‘GGL’, ‘TGG’, and ‘NGE’ had higher contact for semiconducting CNTs.

Furthermore, we performed analysis of the conformational energies of angles and dihedrals for the whole tripeptide and the sidechains of the flanking residues to understand the role of angles and dihedrals to dictate the affinity of tripeptides toward a particular CNT. Based on the results from Figure 5, the conformational energies of angles for both the tripeptides and sidechains of flanking residues were very similar for both metallic and semiconducting CNTs. When looking into the dihedral energies, ‘GGA’, ‘GGL’, ‘TGL’, and ‘TGG’ preferred metallic CNT. Although ‘DGY’, ‘TGY’, and ‘NGE’ had similar dihedral energies for the tripeptides, the sidechains of the flanking residues were larger for the semiconducting CNTs.

Intermediate water molecules between the biological macromolecules and nanoparticles play a crucial role in stabilizing the complex, therefore in this study a detailed analysis of the number of water molecules (Figure 6A,B) between the tripeptides and CNTs was performed. Additionally, the number of hydrogen bonds (Figure 6C,D) formed by these intermediary water molecules were quantified. Logically, for a larger number of interfacial water molecules, the number of hydrogen bonds for these molecules also should be large. The tripeptide combination ‘GGL’, ‘TGY’, and ‘TGK’ had a larger number of interfacial water molecules for metallic CNT, while ‘GGL’ had a lesser number of hydrogen bonds for metallic CNT. ‘TGY’, and ‘TGK’ had larger number of hydrogen bonds. Similarly, ‘GGA’, ‘TGL’, ‘DGY’, ‘TGG’, ‘DGD’, and ‘NGE’ had a larger number of interfacial water molecules for semiconducting CNTs, the number of hydrogen bonds are lesser, except for ‘TGL’. This seems to indicate that the ability of the interfacial water molecules to form hydrogen bonds may have a direct relationship with the interaction energy and stability profile of the two interacting molecules.

## 4. Discussion

To choose a tripeptide combination for selectively sorting metallic or semiconducting CNTs, it must fulfill both the criteria of stability (lower average RMSD) and greater interaction energies at the same time. Based on these criteria, the combinations ‘DGY’ and ‘TGK’ indicate that they are both more stable and have a higher interaction energy for metallic CNTs, making these combinations better for sorting metallic CNTs. Similarly, the tripeptides ‘GGL’, ‘TGG’, and ‘NGE’ demonstrate a better stability and higher interaction energy in the presence of semiconducting CNTs, indicating that these can sort semiconducting CNTs better.

Based on the exhaustive results obtained from the perspective of flanking residues with stability profiles of the two interacting molecules, with glycine as the middle residue, the conditions for sorting metallic and semiconducting CNTs are combined. It is found that these conditions are directly dependent on the type of flanking residue and its influence on the middle glycine residue. There are three distinct cases that are seen when all the results are compared. 

The first scenario is when one of the flanking residues is either a polar-uncharged or positively charged amino acid. The tripeptide ‘TGK’ is stable for metallic CNT and had higher interaction energy (Figure 2B,D). This result agreed with the RMSD and interaction energy of the sidechain (Figure 3B,D). The other combinations containing threonine ‘TGL’ and ‘TGY’ were more stable for semiconducting CNT while exhibiting higher interaction with metallic CNT, due to the tripeptides having higher contact area (Figure 2A,C). On the other hand, ‘TGG’ had lesser deviation in RMSD and greater interaction energy for semiconducting CNT (Figure 2B,D). From these observations, the tripeptide combination ‘TGK’ is more suitable for sorting metallic CNTs. 

In the second case, one of the flanking residues is also glycine, the tripeptide binds equally with metallic or semiconducting CNT because it shows a higher stability in the presence of metallic CNT but also higher interaction energy for semiconducting CNT. This in turn is confirmed from Figure 3A,C, where the glycine, especially in the case of ‘GGL’ and ‘GGA’, goes too close to semiconducting CNT, giving rise to a higher interaction energy, and yet is more stable in the presence of metallic CNT. ‘TGG’ is an interesting case where both the flanking residues glycine and threonine are present in the same tripeptide. It demonstrates higher interaction energy and has more stability with semiconducting CNT (Figure 3B,D). Hence, some cases, such as glycine being on the flanking residue, prompted a further need to understand the affinity of the tripeptides with metallic or semiconducting CNTs. As observed from Figure 6, it is the average number of hydrogen bonds of the interfacial water molecules that ultimately decides if a particular tripeptide combination will favor a metallic or semiconducting CNT, even though ‘GGL’ has a more stable interaction with metallic CNT, the interfacial water molecules form larger number of hydrogen bonds when it interacts with semiconducting CNT. ‘TGG’ on the other hand, even though it has a higher interaction energy with semiconducting CNT, the interfacial water molecules form larger number of hydrogen bonds when ‘TGG’ interacts with metallic CNT. This indicates the role of threonine in sorting metallic CNTs and it proves that glycine (the middle residue), being uncertain in its affinity to either hydrophobic or hydrophilic molecules, switches to polar molecules when flanked by threonine. From Figure 2C,D, with glycine as one of the flanking residues, the tripeptide will select semiconducting CNT if the other flanking residue is hydrophobic (‘GGA’, ‘GGL’) or metallic CNT if the other residue is polar (‘TGG’).

The third scenario is when one of the flanking residues is negatively charged. The tripeptide combination with aspartic acid does seem to favor metallic CNT when the second residue is hydrophobic (‘DGY’) (Figure 2A,C). However, when the second residue is non-hydrophobic (‘DGD’), it had similar interaction energy (Figure 2B,D) and contact area (Figure 3D), but had higher stability towards semiconducting CNT. This is also observed from Appendix A showing the center of mass deviations being more stable for semiconducting compared to metallic CNT. The other tripeptide combination with polar and negatively charged residues (‘NGE’) also selectively binds with the semiconducting CNT. Although the individual amino acids were more stable for metallic CNTs (Figure 4B), the tripeptide (Figure 2B,D) and flanking residues (Figure 3B,D) had higher stability, and interaction energy is higher for semiconducting CNT. 

## 5. Conclusions

Based on the extensive and categorized data analysis of the MD simulations between tripeptides and CNTs (both metallic and semiconducting), we can conclude that the flanking residues play a critical and major role in deciding the affinity towards a type of CNT. From the observed cases, threonine with positively charged lysine as flanking residues in the tripeptide is stable with metallic CNTs, whereas glycine as a flanking residue is versatile in how it interacts with both metallic and semiconducting CNTs. The presence of negatively charged aspartic acid as a flanking residue favors metallic CNTs, and the complimentary residue was hydrophobic, whereas when the other reside was polar, it favored semiconducting CNT. Similarly, the tripeptide with negatively charged and polar flanking residues ‘NGE’ binds with semiconducting CNT despite the individual amino acid being stable for metallic CNT. The observed interaction affinity for CNTs can be used for the sorting of CNTs that can be subsequently used in nanoelectronics and other biotechnology applications, such as drug delivery, protein delivery, and diagnostic biosensor analysis.

## Figures and Tables

**Figure 1 biomolecules-13-00175-f001:**
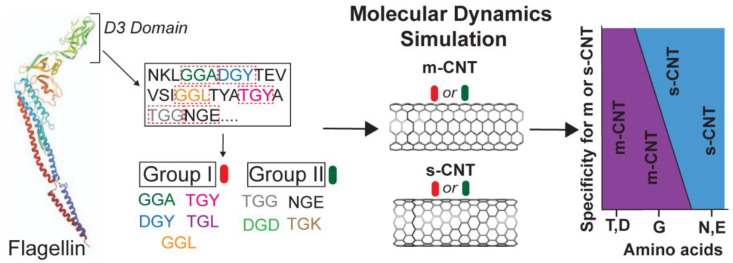
The tripeptide library modelled from the D3 domain of flagellin is composed of nine tripeptides as shown. Each tripeptide has a middle glycine (outlined) as a common denominator with different amino acids flanking on either side. The nine tripeptides are grouped into two categories based on their affinity to water molecules: Group I has at least one hydrophobic flanking residue while Group II has no hydrophobic flanking residues. The specificity of these amino acids towards the CNT types depends on their physical properties which are given in Appendix A.

**Figure 2 biomolecules-13-00175-f002:**
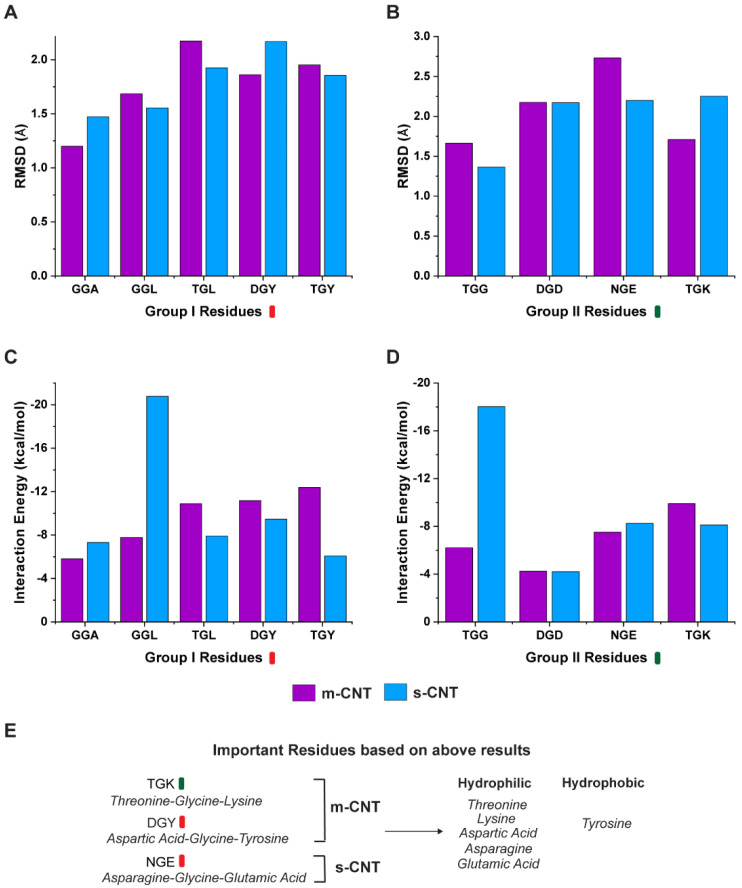
Change in average RMSD per frame for the tripeptide combinations in (**A**) Group I and (**B**) Group II with respect to the metallic and semiconducting CNTs. (**C**,**D**) show the average interaction energies per frame after the stabilization of RMSD for the two groups with respect to both CNTs. (**E**) The important residues based on the results of RMSD and interaction energy of the whole tripeptides with CNTs.

**Figure 3 biomolecules-13-00175-f003:**
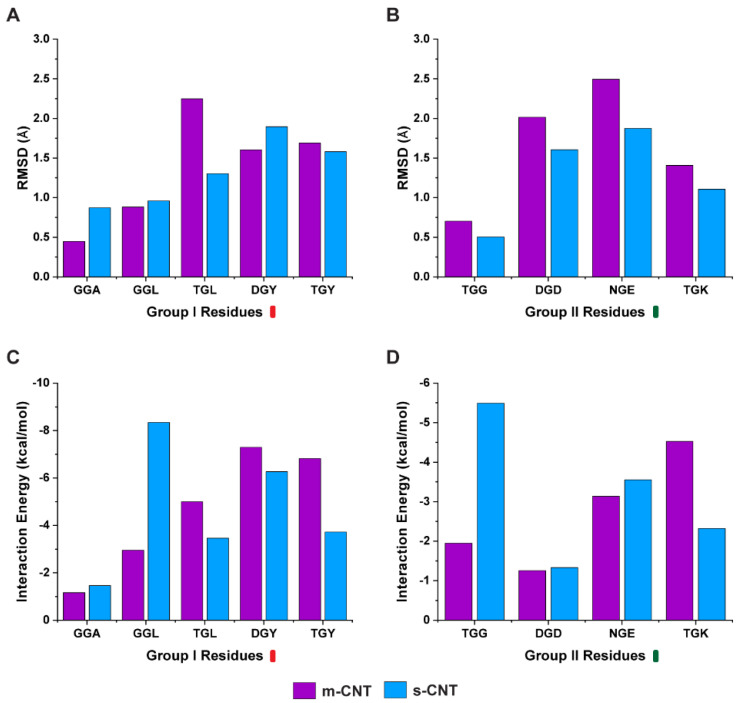
(**A**,**B**) Change in average RMSD for the sidechains of the flanking residues and the (**C**,**D**) interaction energy between the sidechains of the flanking residues and the CNTs for Group I and II tripeptide combinations.

**Figure 4 biomolecules-13-00175-f004:**
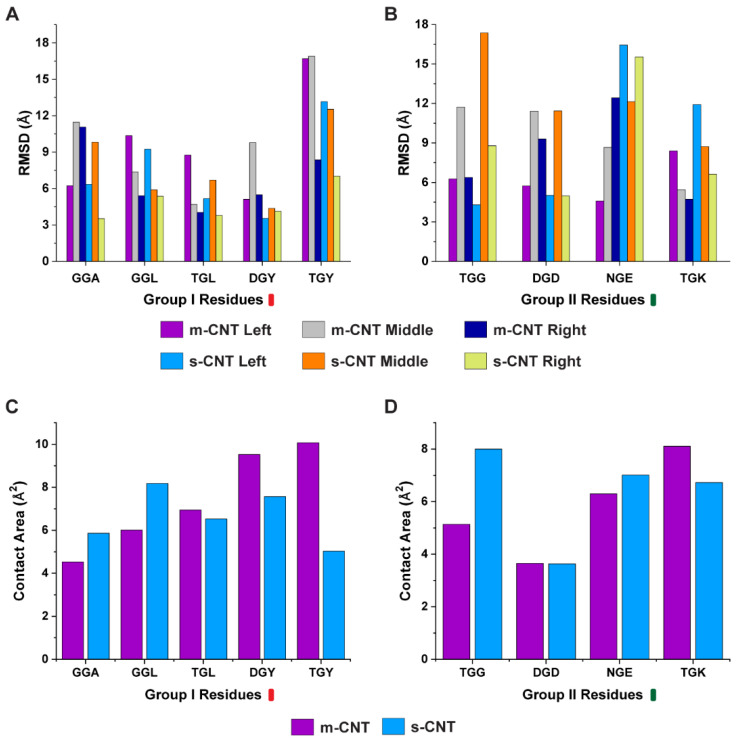
Individual average RMSD analysis for the left (L), middle (M), and right (R) residues of the tripeptides with respect to the metallic and semiconducting CNTs for (**A**) Group I and (**B**) Group II set of tripeptides. (**C**,**D**) show the contact area of the two groups of tripeptide combinations with the metallic and semiconducting CNT.

**Figure 5 biomolecules-13-00175-f005:**
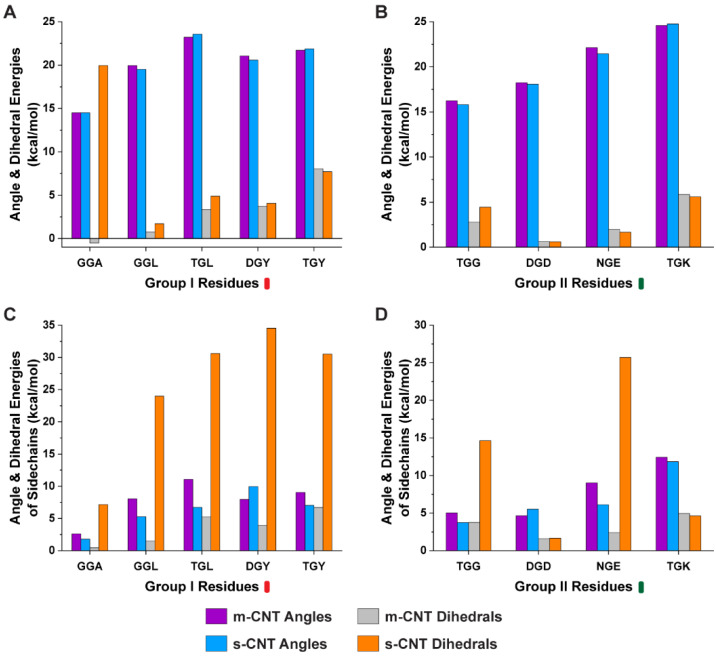
Conformational energies in terms of angle and dihedral energies for (**A**,**B**) tripeptide combinations and (**C**,**D**) sidechains of the tripeptides for groups I and II with respect to metallic and semiconducting CNTs.

**Figure 6 biomolecules-13-00175-f006:**
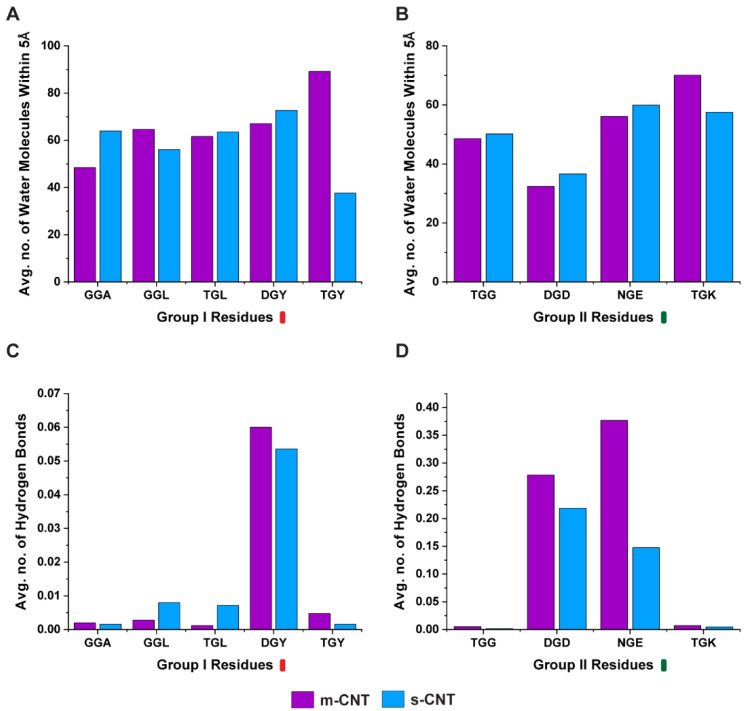
Average number of water molecules within 5Å of the tripeptide for (**A**) Group I and (**B**) Group II combinations. Average number of hydrogen bonds formed by the intermediary water molecules between tripeptide and CNT for (**C**) Group I and (**D**) Group II combinations.

## Data Availability

Modeling and Simulation software, VMD (Visual Molecular Dynamics) and NAMD (Nanoscale Molecular Dynamics) used in this study are freely available from the Theoretical and Computational Biophysics group at the NIH Center for Macromolecular Modeling and Bioinformatics at the University of Illinois at Urbana-Champaign (http://www.ks.uiuc.edu/Research/vmd/—accessed on 1 September 2019). Atomic coordinate (protein data bank, PDB) file for the R-type flagellin filament, 1UCU was obtained from the database located at Research Collaboratory for Structural Bioinformatics (RCSB), www.rcsb.org (accessed on 1 September 2019). To create the individual nine tripeptides, VMD was used to extract the respective tripeptides from the 1UCU PDB file. To create the protein structure files (PSF) and to carry out all-atom simulations, the necessary topology and force field parameter files were obtained from the Chemistry at Harvard Macromolecular Mechanics (CHARMM) database located at the MacKerell Lab at the University of Maryland, School of Pharmacy (http://mackerell.umaryland.edu/charmm_ff.shtml—accessed on 1 September 2019).

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
