# Peer review of "Computational Investigation of Chirality-Based Separation of Carbon Nanotubes Using Tripeptide Library"

_biomolecules, 2023, doi:10.3390/biom13010175_

Round 1
Reviewer 1 Report
This article describes MD simulations between tripeptides and CNTs with (12, 12) and (5, 15) chiralities.
In p2 L58,
To clear the claims of authors, "Specific role of individual amino acids was identified in the tripeptides during their selective interactions with either metallic or semiconducting CNTs", following Q1 and Q2 should be solved.
Q1: Authors calculated for 2 CNTs of (12, 12) and (5, 15) CNTs. At least further 2 CNTs of semiconducting and metallic CNTs must be evaluated to confirm the difference between semiconducting and metallic CNTs.
In L126,
"This provided a clear and compelling reasoning about the nature of the interface between these tripeptides and CNTs, both metallic and semiconducting.", while authors write a number of possible reasons for them in L121-L125.
Q2: What kind of the physical or chemical properties and these differences between "semiconducting" and "metallic" CNT affected on the difference of affinity of tripeptides?
In L264, " ... the ability of the interfacial water molecules to form hydrogen bonds may have a direct relationship with the interaction energy and stability profile of the two interacting molecules."
Q3: Did authors confirmed the difference of (12, 12) and (5, 15) CNTs in the pure water systems?
In L304-L309,
it is confused why non systematic '"NGE' perhaps the most stable middle residue case" was appeared in the last part of discussion, while systematically compared results were discussed before L308. Please move this part to corresponding part of the discussion to clear the discussion.
Other possible modifications.
In Abstract, background of this research should be moved to body text
": Carbon nanotubes (CNT) have fascinating applications in flexible electronics, biosensors, 8 and energy storage devices and are classified as metallic or semiconducting based on their chirality. 9 Semiconducting CNTs have been teased as a new material for building blocks in electronic devices, 10 owing to their band gap resembling silicon. However, CNTs must be sorted into metallic and 11 semiconducting for such applications. Formerly, gel chromatography, ultracentrifugation, size 12 exclusion chromatography, and phage display libraries were utilized for sorting CNTs. 13 Nevertheless, these techniques are either expensive or have poor efficiency."
L55, period is required before "MD".
Author Response
This article describes MD simulations between tripeptides and CNTs with (12, 12) and (5, 15) chiralities.
In p2 L58,
To clear the claims of authors, "Specific role of individual amino acids was identified in the tripeptides during their selective interactions with either metallic or semiconducting CNTs", following Q1 and Q2 should be solved.
Q1: Authors calculated for 2 CNTs of (12, 12) and (5, 15) CNTs. At least further 2 CNTs of semiconducting and metallic CNTs must be evaluated to confirm the difference between semiconducting and metallic CNTs.
- We thank the reviewer for this feedback and appreciate their comment. We do realize that different CNTs having different chirality would interact slightly differently with the given tripeptides. However, basing our results solely on the electrical properties of the CNTs, being either metallic or semiconducting should not change the interactions of the given amino acids to a larger extent in our opinion. We have based our CNT models (metallic and semiconducting) on the difference of chiral vectors being either equal to or not equal to a multiple of 3, which is widely accepted in the scientific community. Furthermore, it is challenging to perform these simulations and the respective analysis with different sets of CNTs in the timeframe given to address the comment.
- To accommodate the reviewer’s comment, we have updated the sentence in L126 to reflect our study better and added the following (highlighted in italics) on page 2, lines 72-73:
“Specific role of individual amino acids was identified in the tripeptides during their selective interactions with either metallic (12,12) or semiconducting CNTs (5,15) used in this study.”
In L126,
"This provided a clear and compelling reasoning about the nature of the interface between these tripeptides and CNTs, both metallic and semiconducting.", while authors write a number of possible reasons for them in L121-L125.
Q2: What kind of the physical or chemical properties and these differences between "semiconducting" and "metallic" CNT affected on the difference of affinity of tripeptides?
- We appreciate the feedback of the reviewer. The molecular dynamics simulations we used via NAMD and VMD are solely done for non-bonding interactions as pointed out on page 4, line 159. This implies that the force fields that are utilized carry the information for the bonds, angles, torsions and dihedrals along with Van der Waals and electrostatic energies, thereby creating the overall potential energy spectrum for the system. The differences in physical properties when the different tripeptides interacted with metallic and semiconducting CNTs are evident from the analyses of RMSD, Van der Waals and electrostatic energies combined with hydrogen bonds and interfacial water molecules. The chemical properties on the other hand would require a different set of conditions involving reactive force fields that involve electronic interactions and hence the formation and disintegration of covalent bonds, something that this study doesn’t focus on.
In L264, " ... the ability of the interfacial water molecules to form hydrogen bonds may have a direct relationship with the interaction energy and stability profile of the two interacting molecules."
Q3: Did authors confirmed the difference of (12, 12) and (5, 15) CNTs in the pure water systems?
- We thank the reviewer for this comment. In molecular dynamics simulations, during the first phase of energy minimization and equilibration, a typical goal is to reduce the residual potential energy of the system using a search algorithm such as steepest descent to find the minimum potential energy (global minimum) and record the respective co-ordinates of all the atoms (state of the system). In order to perform this minimization simulation, it is required that the boundary conditions are maintained sufficiently enough to avoid unnecessary vibrations of the atoms. This is taken care of by adding neutralizing salt concentration to the water box. Even though it is possible to perform minimization simulation without salt, the algorithm may not be able to find the global minimum in a given time frame without charges in the form of salt ions. The aim of this study is also to analyze the interactions between the tripeptides and CNTs in water systems and that hydrogen bonds between the water molecules and the tripeptide-CNTs complex might aid in the stability of tripeptide-CNT complex. While we are aware that there can be some hydrogen bonds between (12,12) or (5,15) and water without the tripeptide, we believe that the difference will not be substantial. Even with the tripeptide in the system, the number of hydrogen bonds is not significantly high due to the inherent hydrophobic nature of (12,12) and (5,15). However, we thank the reviewer for this comment and shall keep this comment in mind for future studies.
In L304-L309,
it is confused why non systematic '"NGE' perhaps the most stable middle residue case" was appeared in the last part of discussion, while systematically compared results were discussed before L308. Please move this part to corresponding part of the discussion to clear the discussion.
We thank the reviewer for this constructive comment and have rearranged the discussion section to fit the flow of the narrative.
Other possible modifications.
In Abstract, background of this research should be moved to body text
": Carbon nanotubes (CNT) have fascinating applications in flexible electronics, biosensors, 8 and energy storage devices and are classified as metallic or semiconducting based on their chirality. 9 Semiconducting CNTs have been teased as a new material for building blocks in electronic devices, 10 owing to their band gap resembling silicon. However, CNTs must be sorted into metallic and 11 semiconducting for such applications. Formerly, gel chromatography, ultracentrifugation, size 12 exclusion chromatography, and phage display libraries were utilized for sorting CNTs. 13 Nevertheless, these techniques are either expensive or have poor efficiency."
We thank the reviewer for the suggestion. The background of the techniques mentioned in the abstract has been expanded in the introduction section, page 1-2, lines 36-49.
L55, period is required before “MD”.
We have addressed the concerned sentence error.
Reviewer 2 Report
The authors investigate, by means of molecular dynamics simulations, how the chirality of Carbon Nanotubes affects its interaction energy with nine different tripeptides. Interestingly they found almost all the tripeptides can differentiate between metallic and semiconductors CNT even though the method used for describing the interaction does not take into account the electronic part of the system. The paper, which contains many results, well presented and analyzed, can be considered for publication in the journal. However, before publication, the authors should a) improve the computational description by adding information on how many tripeptides and water molecules they introduced in each simulation and b) correct the caption of the figures since “Fig 6” is missing whereas “Fig.4” appears two times.
Author Response
The authors investigate, by means of molecular dynamics simulations, how the chirality of Carbon Nanotubes affects its interaction energy with nine different tripeptides. Interestingly they found almost all the tripeptides can differentiate between metallic and semiconductors CNT even though the method used for describing the interaction does not take into account the electronic part of the system. The paper, which contains many results, well presented and analyzed, can be considered for publication in the journal. However, before publication, the authors should
- A) improve the computational description by adding information on how many tripeptides and water molecules they introduced in each simulation and
We thank the reviewer for this constructive comment. We have revised the computational description on page 3, lines 107-108 by adding the breakdown in terms of the CNT, tripeptide and number of water atoms for individual simulations.
- b) correct the caption of the figures since “Fig 6” is missing whereas “Fig.4” appears two times.
We thank the reviewer for pointing out the incorrect label in the figure captions. We have corrected the issue.
Round 2
Reviewer 1 Report
Author responded to all of my reviewing comments and modified manuscript based on them. I recommend this manuscript to be published in Biomolecules journal.